# Adverse Effects of Infant Formula Made with Corn-Syrup Solids on the Development of Eating Behaviors in Hispanic Children

**DOI:** 10.3390/nu14051115

**Published:** 2022-03-07

**Authors:** Hailey E. Hampson, Roshonda B. Jones, Paige K. Berger, Jasmine F. Plows, Kelsey A. Schmidt, Tanya L. Alderete, Michael I. Goran

**Affiliations:** 1The Saban Research Institute, Los Angeles, Children’s Hospital, Los Angeles, CA 90027, USA; hhampson@usc.edu (H.E.H.); rbarnerjones@gmail.com (R.B.J.); paberger@chla.usc.edu (P.K.B.); jasmineplows@gmail.com (J.F.P.); kelschmidt@chla.usc.edu (K.A.S.); 2Department of Epidemiology, University of Southern California, Los Angeles, CA 90007, USA; 3Department of Integrative Physiology, University of Colorado Boulder, Boulder, CO 80309, USA; tanya.alderete@colorado.edu

**Keywords:** breastmilk, child eating behavior questionnaire, corn-syrup solids, eating behavior, enjoyment of eating, food fussiness, formula, Hispanic, infant, obesity

## Abstract

Few studies have investigated the influence of infant formulas made with added corn-syrup solids on the development of child eating behaviors. We examined associations of breastmilk (BM), traditional formula (TF), and formula containing corn-syrup solids (CSSF) with changes in eating behaviors over a period of 2 years. Feeding type was assessed at 6 months in 115 mother–infant pairs. Eating behaviors were assessed at 12, 18 and 24 months. Repeated Measures ANCOVA was used to determine changes in eating behaviors over time as a function of feeding type. Food fussiness and enjoyment of food differed between the feeding groups (*p* < 0.05) and changed over time for CSSF and TF (*p* < 0.01). Food fussiness increased from 12 to 18 and 12 to 24 months for CSSF and from 12 to 24 months for TF (*p* < 0.01), while it remained stable for BM. Enjoyment of food decreased from 12 to 24 months for CSSF (*p* < 0.01), while it remained stable for TF and BM. There was an interaction between feeding type and time for food fussiness and enjoyment of food (*p* < 0.01). Our findings suggest that Hispanic infants consuming CSSF may develop greater food fussiness and reduced enjoyment of food in the first 2 years of life compared to BM-fed infants.

## 1. Introduction

Added sugar is widespread in our food system and is characteristic of the Western Style diet. This is a concern as added sugar is associated with obesity development and a wide range of co-morbid conditions [1,2]. Evidence suggests that chronic consumption of added sugar may affect obesity risk as it is consumed in excess and contributes empty calories to the diet [3,4]. However, it is also postulated that early exposure to sugars may indirectly influence obesity risk by shaping taste preferences, self-regulation of energy intake, and the reinforcing value of food [5,6,7,8,9]. These eating behaviors may ultimately lead to an overindulgence of sweet foods and rejection of bitter-tasting alternatives. This is relevant as food manufacturers incorporate sugars in the form of added corn-syrup solids into infant formulas and introductory solids, despite the most recent dietary guidelines recommending zero added sugars in the first 2 years of life [4].

Human studies have shown that the introduction of added sugar in infancy influences eating behavior in childhood. For example, infants fed added sugar before 4 months were more likely to reject bitter and sour-tasting foods in early childhood. These infants were found to be at an increased risk for childhood obesity [5]. Furthermore, infants that are fed formulas with sweet flavors have a lower acceptance of bitter tasting foods, such as broccoli, compared to infants fed formulas with bitter flavors [10]. While the exact biological mechanism has yet to be fully elucidated, animal studies suggest that the gut microbiome may be a key link between early sugar exposure and infant food preferences, leading to later obesity risk [11].

Although several studies have examined the effects of breastfeeding and formula feeding on infant eating behaviors, no study has examined whether infant formula made with corn-syrup solids instead of lactose as the major carbohydrate source may have additional, potentially adverse effects. Various infant formulae have been developed without lactose due to potential concerns of lactose intolerance, some of which use corn syrup solids. Therefore, the aim of this study was to determine associations of breastfeeding, traditional formula, and formula containing corn-syrup solids at 6 months with changes in eating behaviors at and between 12, 18, and 24 months.

## 2. Materials and Methods

### 2.1. Study Participants

We obtained data from the Southern California Mother’s Milk Study, which is an on-going longitudinal cohort study of Hispanic mother-infant dyads, as reported previously [12]. Briefly, participants were eligible if they (a) self-identify as Hispanic (mother and father), (b) had a singleton birth, (c) intended to breastfeed for at least 3 months, (d) enrolled within 1 month of the infant’s birth, and (e) were willing to/had the ability to understand the procedures of the study and be able to read English or Spanish at the fifth-grade level. Participants were not eligible if they (a) had a physician diagnosis of a major medical illness or eating disorder, (b) had a physical, mental or cognitive issue that prevented participation, (c) reported chronic use of medication that may affect body weight or composition, insulin resistance or lipid profiles, (d) were a current smoker or were a user of other recreational drugs, (e) had a pre-term/low birth weight infant or diagnosis of any fetal abnormalities, or (f) were less than 18 years old.

At each study visit, mothers reported health history and information on breastfeeding, which were included as covariates in the analysis [12]. Institutional review boards at the University of Southern California and Children’s Hospital Los Angeles approved the study. Written informed consent was obtained from all mothers prior to data collection. Mother–infant pairs were included in the final analysis if they completed all 6, 12, 18 and 24 month visits (Figure 1).

### 2.2. Infant Feeding Modality Determination

Two 24 h dietary recalls were conducted with mothers over the phone by trained bilingual research personnel. Dietary intake collection and analysis was performed using the Nutritional Data System for Research software (NDSR) [13]. The NDSR data were used to calculate the infants’ breastmilk and formula-type groups. Breastmilk-fed infants were categorized based on predominant consumption type (receiving more than 80% of feedings from breastmilk) at 6 months. Feeding type groups included infants who received primarily human milk directly from the breast or pumped human milk from a bottle (BM, n = 43), infants who received a traditional formula, which was lactose and cow-milk based (TF, n = 41), and infants who received a lactose-reduced formula made with corn syrup solids (CSSF, n = 31), in this case Enfamil Gentlease, Similac Isomil Advance, produced by Mead Johnson & Company, LLC [14,15]. Infants in the TF group received one of the following formulas: Similac Advance 20, Similac Advance EarlyShield, Gerber Good Start Gentle, Enfamil A.R., and Enfamil Infant, made by a variety of formula companies.

### 2.3. Eating Behavior Assessment

To assess child eating behaviors at 12, 18 and 24 months, we used the Children’s Eating Behavior Questionnaire (CEBQ), a maternal report survey tailored to children 12 months of age and older [16,17,18]. The questionnaire was available in English and Spanish. The CEBQ consists of 35 questions organized into 8 domains, i.e., satiety responsiveness, slowness in eating, emotional under-eating, food fussiness, enjoyment of food, food responsiveness, desire to drink, and emotional over-eating. All questions were ranked on a Likert scale of Never (0) to Always (4), and scores were averaged within domains [16,17,18].

### 2.4. Participant Characteristics

We collected anthropometrics and sociodemographic information, including maternal pre-pregnancy BMI (kg/m^2^) (determined from height and weight recall prior to pregnancy), infant sex, infant age (days), infant birth weight (kg), mode of delivery (vaginal vs. Caesarean Section), and socioeconomic status index (SES). We determined socioeconomic status using The Hollingshead Index. Students, stay-at-home parents, and unemployed persons do not have assigned employment categories in the Hollingshead Index, therefore, a score of zero was given to these individuals under the assumption they these participants likely to have very little or no income. This was done in order to retain these participants in the analyses.

### 2.5. Statistical Analysis

Basic descriptive statistics were calculated for infant feeding modality, eating behavior domains, and all covariates included in the analysis. One-Way Analysis of Covariance (ANCOVA) was performed to determine initial differences in covariates between feeding-type groups. Model diagnostics were performed and any variables not meeting assumptions of ANCOVA were transformed using the bestNormalize package in R, which selects the best method to normalize the variable [19]. One-Way ANCOVA was performed at each time point to determine differences in eating behaviors by infant feeding modality. Repeated-Measures ANCOVA was conducted including an interaction term between time (infant age) and infant feeding modality at 6 months, to determine differences in eating behaviors over time as a function of infant feeding modality. For significant between-group differences and interactions, post hoc pairwise comparisons using Tukey’s Test were performed. Based on the literature, we adjusted for the following common covariates: maternal pre-pregnancy BMI (kg/m^2^), infant breast milk feedings and formula feedings per day (in a ratio variable of breastmilk feedings to formula feedings normalized across all children in the sample), infant sex, infant age (days), infant birth weight (kg), mode of delivery, and socioeconomic status index (SES). We performed reliability testing on CEBQ domain summary scores at all timepoints, 12, 18 and 24 months, using the Psych package in R and an inclusion alpha level of 0.7 [20]. Domains not meeting the significance level 0.7 were removed from analysis. RStudio was used for all analyses using R version 4.0.5 [21]. We performed a sensitivity analysis to determine if results changed when we distinguished between two types of breastmilk feeding. We divided breastmilk-fed infants into two groups, those directly breastfed (BF), and those fed breastmilk pumped in a bottle (PB). We ran all statistical analyses with all four groups, TF, CSSF, BF and PB, and results did not differ from the original analysis. Therefore, we collapsed the BF and PB groups into one breastmilk group (BM).

## 3. Results

One-hundred and fifteen mother–infant pairs were included in this study. Fifty-six percent of infants were female, and the mean infant birth weight was 3.4 ± 0.4 kg. The mean breastmilk feedings per day at 6 months was 2.7 ± 3.4 and the mean formula feedings per day was 2.9 ± 3.0 while the number of feedings per day was not significantly different between breastfed and formula-fed infants. Initial maternal and infant characteristics were similar across groups (Table 1). Twenty-seven percent of infants were in the CSSF group, 36% in the TF group, and 37% in the BM group. Cronbach’s alpha analysis for internal consistency of CEBQ domains suggested that the domains satiety responsiveness (a = 0.61–0.63) and slowness in eating (a = 0.46–0.49) did not have a high enough reliability above a = 0.70. An analysis was not performed on these domains. All other CEBQ domains had an acceptable alpha level above 0.7 (0.71–0.83).

Food fussiness was significantly different between infant feeding groups (*p* = 0.01). For the CSSF group, food fussiness significant increased from 12 to 18 months (mean increase = 0.51, *p* < 0.01), and 12 to 24 months (mean increase = 0.77, *p* < 0.001) (Figure 2). Food fussiness did not change significantly over time for the other groups. There was a significant interaction between infant feeding modality at 6 months and time (infant age) in months for food fussiness (*p* < 0.01) (Table 2). In post hoc testing, food fussiness was significantly greater in the CSSF group compared to the BM group at 24 months (mean difference = 0.54, *p* < 0.05).

Enjoyment of food was significantly different between infant feeding groups (*p* = 0.001). For the CSSF group, enjoyment of food significantly decreased from 12 to 24 months (mean decrease = 0.60, *p* < 0.001), while it remained constant in the other groups over time (Figure 3). Furthermore, there was a significant interaction between infant feeding modality at 6 months and time (infant age) in months for enjoyment of food (*p* = 0.002) (Table 2).

## 4. Discussion

In this study, we found that the development of child eating behaviors in the first 2 years of life was affected by early exposure to formula made with corn-syrup solids as compared to traditional formula made with lactose and compared to breastmilk. Specifically, early exposure to formula made with corn-syrup solids was associated with increased food fussiness and reduced enjoyment of food over time. In addition, CSSF-fed infants had worsening eating behaviors over time, compared to TF-fed and BM-fed infants. Importantly, the behaviors observed in the CSSF-fed infants have been linked to poor diet quality and variety, which in turn are associated with obesity and related co-morbidities in childhood [7,22,23].

Our findings provide additional insights into the early exposure to corn-syrup solids and infant eating behaviors examined in prior studies. For example, Shepard et al. found that infants fed “gentle” formula containing corn-syrup solids demonstrated consistent eating behaviors from 3 to 5 months, in contrast to our findings that CSSF-fed infants exhibited changes in eating behaviors over a longer period of time [24]. Shepard et al. examined CSSF intake at an earlier period than the present study and included fewer formula type groups, which may explain differences in findings. Because eating behaviors emerge over time, it is plausible that the prior study did not have a long enough duration so as to capture changes in eating behavior as we did. Consistent with our findings, Murray et al. reported that infants who were exposed to sweet, non-milk solids and beverages had decreased bitter food acceptance and increased obesity risk, a trademark of picky eating behavior [5]. Similarly, we found that CSSF-fed infants had a greater increase in food fussiness over time, relative to formula fed infants and breastfed infants. It may be that early exposure to added sugar in the form of corn-syrup solids enhances the infant’s affinity for sweet tastes and exacerbates innate disliking of bitter tastes, which may contribute to picky eating [5,7,10].

This hypothesis is supported by work from Mennella and colleagues, who found that infants fed sweet-tasting formula were more likely to reject broccoli than those who had exposure to bitter-tasting formula [10]. Similarly, we found that all feeding groups had a similar degree of food fussiness at 12 months of age. However, we found that there was a greater increase in food fussiness among CSSF-fed infants compared to TF-fed infants, and food fussiness remained stable among BM-fed infants. This suggests that infants exposed to diverse flavors or diminished sweet flavors in breastmilk or traditional formula are more likely to accept wide-ranging foods compared to those exposed to the sweetness of CSSF. These experiences may also influence enjoyment of food among CSSF-fed infants, as picky eating behaviors may reduce overall enjoyment of a diverse diet that comes with the introduction of solid foods [23,25].

While several mechanisms may underlie our findings, a potential explanation that has gathered the most interest is the effects of early sugar exposure on the infant gut microbiome, which may affect appetite regulation through gut-derived hormones [11,12,26,27,28]. Simple carbohydrates and other dietary factors can serve as energy for a harmful subset of gut microbes, which may ultimately displace beneficial bacteria [26,27,28]. The increased proportions of harmful microbes therefore receive more energy from the host, which has a strong influence on appetite, metabolism and food-related behaviors [28]. For example, harmful bacteria may ferment added sugar and produce metabolites, primarily short-chain fatty acids, which can directly affect hunger and signaling [27]. Through this mechanism, CSSF-fed infants may have an increased affinity for sweet-tasting foods. As infants are introduced to more solid foods with a range of flavors, they may experience a decreased preference for and enjoyment of food with bitter tastes, contributing to food fussiness [23].

In addition, animal studies have shown that rats with early sugar exposure exhibit addiction-like responses, with an increased affinity for added sugars and symptoms of withdrawal [29]. These patterns are linked to opioid-receptor binding associated with the added sugar intake and neurochemical changes that result in increased dopamine from sugar consumption [29]. Early added corn-syrup exposure from CSSF could program an infant for a heighted dopamine response that is not met with the introduction of bitter foods, thus reducing their enjoyment of food. The potential effects on brain-based eating behaviors may also be exacerbated by displacement of lactose with corn-syrup solids in the CSSF. Decreased exposure to lactose and therefore, galactose, may also have deleterious effects on infant-brain development and subsequent healthy eating behavior in children [12,14,15,30].

This study has several limitations. We used a self-report survey to assess child eating behaviors, which could result in response bias. However, we only used CEBQ domains with high validity. It is possible that the translation of the CEBQ into Spanish may have affected the validity of the questionnaire. Further, this is an observational study, and we cannot make causal conclusions based on our findings. However, our study was longitudinal in nature, which adds a further temporal component to the findings. Another important limitation is that we do not have information on why mothers chose different infant formulas. However, our sample has no significant differences in maternal or infant baseline characteristics between groups, including socioeconomic status, which may be influential for selection of infant formula type. Additionally, one of the infant formulas categorized in the TF group due to its lactose and cow’s milk content is Gerber Good Start Gentle. Similar to Enfamil Gentlease in the CSSF group, this product is marketed toward colicky and fussy infants. A systematic review by Belamarich et al. reports that mothers choose these gentle formulas due to the marketing strategies that imply crying and fussiness are indicative of digestive discomfort [31]. This sheds some light on the choices that mothers may be making with regard to formula, however, given that there are gentle formulas included in both groups, we do not suspect that this contributes to, nor fully explains the differences seen in our study. However, maternal beliefs related to formula choice is an important area for further study.

These findings may have important implications for infant feeding recommendations given that early life added corn-syrup exposure may influence taste-preference formation, food rejection with the introduction of solids, and the composition of the infant gut microbiome. Food fussiness and the reduced enjoyment of food may inhibit the acceptance of fruits and vegetables and other healthful foods in the diet, leading to poor diet quality and variety. Importantly, CSSF is only consumed by <10% of infants in the general US population [15]. However, in our cohort of Hispanic mothers, 50% of formula-fed infants consume CSSF. Therefore, the results of this study are highly relevant for this subset of the population that is already at higher risk for obesity and related chronic disease development.

## 5. Conclusions

These findings suggests that Hispanic infants who receive formula made with corn syrup solids in place of lactose develop poor eating behavior in the first 2 years of life, including greater food fussiness and reduced enjoyment of food, compared to traditional formula-fed and breastmilk-fed infants. Further studies are needed to elucidate the underlying mechanisms by which added corn-syrup solids influence child eating behaviors as well as clinical health outcomes. Our findings provide evidence for future studies exploring the effects of infant formulas with reduced lactose and added corn-syrup solids on children’s eating behaviors and growth.

## Figures and Tables

**Figure 1 nutrients-14-01115-f001:**
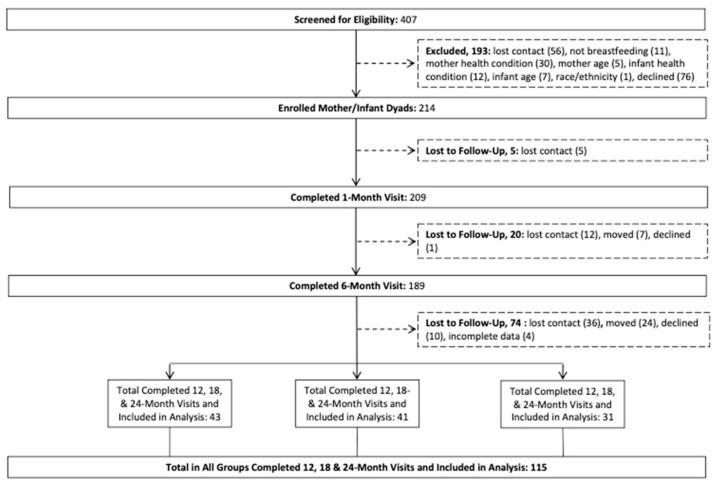
Participation flow chart and distribution of participants to infant feeding type groups.

**Figure 2 nutrients-14-01115-f002:**
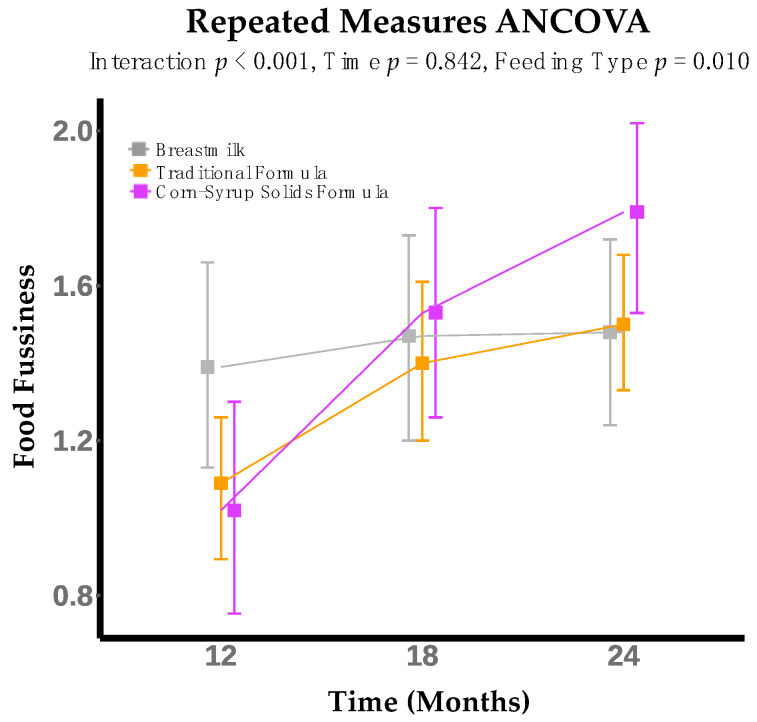
Food Fussiness increases from 12 to 18 months and 18 to 24 months for infants who consumed formula with added corn-syrup solids (CSSF) at 6 months. Food fussiness increases from 12 to 24 months for infants who consume traditional infant formula (TF) at 6 months. The model was adjusted for maternal pre-pregnancy BMI (kg/m^2^), infant breast milk feedings per day and formula feedings per day, infant sex, infant birth weight (kg), mode of delivery, and socioeconomic status index (SES).

**Figure 3 nutrients-14-01115-f003:**
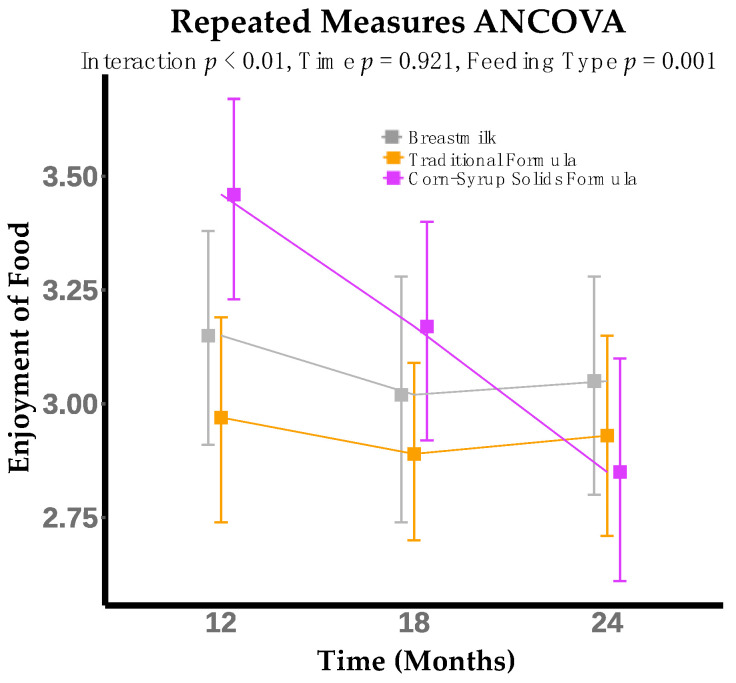
Enjoyment of food decreases from 12 to 24 months for infants who consumed formula with added corn-syrup solids (CSSF) at 6 months. Model adjusted for maternal pre-pregnancy BMI (kg/m^2^), infant breast milk feedings per day and formula feedings per day, infant sex, infant birth weight (kg), mode of delivery, and socioeconomic status index (SES).

**Table 1 nutrients-14-01115-t001:** Descriptive statistics based on feeding type group at 6 months (n = 115). No significant differences were found in maternal or infant characteristics between groups based on One-Way ANOVA.

Participant Characteristics	Analysis of Variance			
	(Between-Group Differences by Feeding Type at 6 Months)			
	BM *	TF *	CSSF *	*p*-Values
Infant Birth Weight (mean, kg)	3.37	3.42	3.40	0.840
Socioeconomic Status Index	25.40	26.10	27.00	0.850
Infant Sex (% Female)	58%	46%	65%	0.290
Mode of Delivery (%Vaginal)	81%	76%	71%	0.580
Maternal Pre-Pregnancy BMI (mean, kg/m^2^)	27.20	29.40	27.50	0.140

* CSSF = Formula with Reduced-Lactose, Added-Corn Syrup Solids, BM = Breastmilk, TF = Traditional Formula.

**Table 2 nutrients-14-01115-t002:** Repeated measures ANCOVA shows that there was a statistically significant interaction between feeding modality and time from 12 to 24 months for food fussiness and enjoyment of food. Model includes the covariates maternal pre-pregnancy BMI (kg/m^2^), infant breast milk feedings per day, infant formula feedings per day, infant sex, infant birth weight (kg), mode of delivery, and socioeconomic status index (SES).

Eating Behaviors	Time	Feeding Type	Interaction ^a^
Food Responsiveness	0.902	0.727	0.748
Food Fussiness	0.187	0.109	0.004 **
Enjoyment of Food	0.388	0.002 **	0.001 **
Desire to Drink	0.005 **	0.765	0.409
Emotional Undereating	0.732	0.860	0.989
Emotional Overeating	0.391	0.484	0.418

^a^ Interaction between Time and Feeding Type. *p* < 0.01 **.

## Data Availability

Data described in the manuscript, code book, and analytic code will be made available upon request. Data requests can be made to Michael I. Goran, (323) 217–5116, goran@usc.edu.

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
