# Peer review of "Adverse Effects of Infant Formula Made with Corn-Syrup Solids on the Development of Eating Behaviors in Hispanic Children"

_nutrients, 2022, doi:10.3390/nu14051115_

Round 1

Reviewer 1 Report

Overall, a nice and well-written paper on an important topic. The impact of early-childhood feeding on later eating behavior is an important topic that is currently developing more and more. The findings that Hampson and co-authors present in this work form an important addition to the field.

Couple of points that require some attention:

-the authors disclose the CSS formula (Gentlease) but not the traditional formula. I believe both should be disclosed

-when considering the Gentlease formula, the use of CSS is one distinction from traditional formula, but the second is that it used hydrolysed milk proteins, not intact ones like present in human milk and traditional formula. The authors should discuss if this could affect outcomes

-I’m missing any discussion on why mothers chose to give their infants certain formula. Particularly Gentlease is marketed towards infants with some digestive discomfort. So is there a stratification happening already in the groups that goes beyond formula being used? This should be discussed

-Would be good to see some insights into the feeding behavior. The key stratification point is the 6 month mark, upon which classifications are done. But what was feeding behavior after this? How long were children on formula or breast milk? This can be an important factor that is currently not clear.

I feel these points need to be discussed to highlight the true value of the outcomes.

Author Response

Manuscript ID Number: nutrients-1598027

Thank you for your consideration of our manuscript for publication in Nutrients. My co-authors and I appreciate the thorough evaluation provided by the reviewers. We have addressed concerns from reviewer 1 below, with corresponding changes in the text.

Reviewer 1

  1. The authors disclose the CSS formula (Gentlease) but not the traditional formula. I believe both should be disclosed.

Response: We have modified the text to include the list of traditional cow’s milk- and lactose-based formulas (lines 93-95): “Infants in the TF group received one of the following formulas, Similac Advance 20, Similac Advance EarlyShield, Gerber Good Start Gentle, Enfamil A.R., and Enfamil Infant, made by a variety of formula companies.” Moreover, there was an oversight in our initial submission regarding the commercial formulas that were categorized as CSSF. In addition to Enfamil Gentlease, Similac Isomil Advance was included, which contains added corn syrup solids as the main ingredient and was also used to define the CSSF group. This information has been added to the text.

  1. When considering the Gentlease formula, the use of CSS is one distinction from traditional formula, but the second is that it used hydrolysed milk proteins, not intact ones like present in human milk and traditional formula. The authors should discuss if this could affect outcomes.

Response: The reviewer highlights an important point that the Gentleease formula included in the CSSF group contains hydrolyzed proteins rather than intact protein. However, the presence of hydrolyzed proteins is not unique to the CSSF group. For example, Gerber Good Start Gentle formula also contains the hydrolyzed proteins. Moreover, Similac Isomile Advance in the CSSF group as well as the remaining formulas within the TF group all contain intact protein. It follows that the main distinction between CSSF and TF groups is the presence of added corn syrup solids rather than hydrolyzed protein. Therefore, we hypothesized that exposure to the added corn syrup solids underlies the differences we observed in eating behavior between the different formula groups.

  1. I’m missing any discussion on why mothers chose to give their infants certain formula. Particularly Gentlease is marketed towards infants with some digestive discomfort. So is there a stratification happening already in the groups that goes beyond formula being used? This should be discussed.

Response: We do not have information on why mothers chose different infant formula types. However, we found that there were no significant differences in maternal or infant baseline characteristics between groups, including socioeconomic status, which may influence selection of infant formula type. Furthermore, one of the infant formulas categorized as traditional formula (TF) due to its lactose and cow’s milk content is Gerber Good Start Gentle. This product is similarly marketed toward colicky and fussy infants. While a limitation of our study is that we do not have data on maternal beliefs related to dietary components, including sugar and lactose, there is likely not a systematic difference in terms of the gastrointestinal composition of infants in the TF group vs. the CSSF group. A systematic review by Belamarich et al. reports that mothers choose gentle formulas due to the marketing strategies of these formulas that imply crying and fussiness are indicative of digestive discomfort [1]. This sheds some light on the choices that mothers may be making with regard to formula, but again, given that there are gentle formulas included in both groups, we do not suspect that this contributes to nor fully explains the differences seen in our study. However, maternal beliefs related to formula choice is an important area for further study. We have included this discussion of formula choice in the manuscript (lines 270-283).

  1. Would be good to see some insights into the feeding behavior. The key stratification point is the 6-month mark, upon which classifications are done. But what was feeding behavior after this? How long were children on formula or breast milk? This can be an important factor that is currently not clear.

Response: We chose to study the 6-month timepoint specifically because it captures the period when infants have been exclusively fed human milk or formula. If we were to include categorization after the 6-month timepoint, it would be difficult to tease out the effects of solid foods that are introduced at this time point. Instead, we focus on the important early time window where infants have most rapid growth and development and there is not confounding by solid foods. Furthermore, to adjust potential differences in formula and breastmilk feeding from the exposure timepoint at 6 months, throughout the study period we control for formula and breastfeeding at each time point of study (12, 18 and 24) by including a breastmilk-formula ratio variable in our statistical model that adjusts for feedings at each timepoint.

References

  1. Belamarich, P.F.; Bochner, R.E.; Racine, A.D. A Critical Review of the Marketing Claims of Infant Formula Products in the United States. Clin Pediatr (Phila) 2016, 55, 437–442, doi:10.1177/0009922815589913.

Reviewer 2 Report

I suggest to explain more strongly the limits of the study (i.e. data only in hispanic population it's a possible bias of selection, explain the reason why you choose only Hispanic people)

in statistical analysis explain the category of socio-economical status (according to which scale was done)

finally I suggest to put the flow-chart of the study in the main text not as supplementary material

Author Response

 Manuscript ID Number: nutrients-1598027

Thank you again for your consideration of our manuscript for publication in Nutrients. My co-authors and I appreciate the constructive feedback provided by reviewer 2. We have addressed their concerns below, with corresponding changes in the text.

Reviewer 2

  1. I suggest to explain more strongly the limits of the study (i.e. data only in Hispanic population it's a possible bias of selection, explain the reason why you choose only Hispanic people)

Response: We understand the reviewer’s concern. This study focused on Hispanic mother-infant pairs residing in Los Angeles County because these families are vulnerable to health disparities that influence risk for obesity, which may be evident in the first 24 months of life. Therefore, our aim is to generalize our results to Hispanic mothers and infants in LA country. We do not aim to generalize our results to all Hispanic mothers and infants, nor to other racial/ethnic groups. Given that selection bias is introduced when a sample is not representative of the target population, we do not likely have selection bias in this case.

  1. In statistical analysis explain the category of socio-economical status (according to which scale was done)

Response: We have added this information to the text in lines 108-113.

  1. Finally, I suggest to put the flow-chart of the study in the main text not as supplementary material.

Response: The flow-chart has been included as a component of the main text.

Round 2

Reviewer 1 Report

The comments I raised have been addressed adequately